# Depleting Ly6G Positive Myeloid Cells Reduces Pancreatic Cancer-Induced Skeletal Muscle Atrophy

**DOI:** 10.3390/cells11121893

**Published:** 2022-06-10

**Authors:** Michael R. Deyhle, Chandler S. Callaway, Daria Neyroud, Andrew C. D’Lugos, Sarah M. Judge, Andrew R. Judge

**Affiliations:** 1Department of Physical Therapy, University of Florida, Gainesville, FL 32610, USA; mdeyhle@unm.edu (M.R.D.); chandlercallaway@phhp.ufl.edu (C.S.C.); daria.neyroud@me.com (D.N.); adlugos@phhp.ufl.edu (A.C.D.); smsenf@phhp.ufl.edu (S.M.J.); 2Department of Health, Exercise & Sports Sciences, University of New Mexico, Albuquerque, NM 87131, USA; 3Faculty of Biology and Medicine, Institute of Sport Sciences, University of Lausanne, Quartier UNIL-Centre, Building Synathlon, 1015 Lausanne, Switzerland

**Keywords:** skeletal muscle, cachexia, MDSC, atrophy, immunosuppression

## Abstract

Immune cells can mount desirable anti-cancer immunity. However, some immune cells can support cancer disease progression. The presence of cancer can lead to production of immature myeloid cells from the bone marrow known as myeloid-derived suppressor cells (MDSCs). The immunosuppressive and pro-tumorigenic effects of MDSCs are well understood. Whether MDSCs are involved in promoting cancer cachexia is not well understood. We orthotopically injected the pancreas of mice with KPC cells or PBS. One group of tumor-bearing mice was treated with an anti-Ly6G antibody that depletes granulocytic MDSCs and neutrophils; the other received a control antibody. Anti-Ly6G treatment delayed body mass loss, reduced *tibialis anterior* (TA) muscle wasting, abolished TA muscle fiber atrophy, reduced diaphragm muscle fiber atrophy of type IIb and IIx fibers, and reduced atrophic gene expression in the TA muscles. Anti-ly6G treatment resulted in greater than 50% Ly6G+ cell depletion efficiency in the tumors and TA muscles. These data show that, in the orthotopic KPC model, anti-Ly6G treatment reduces the number of Ly6G+ cells in the tumor and skeletal muscle and reduces skeletal muscle atrophy. These data implicate Ly6G+ cells, including granulocytic MDSCs and neutrophils, as possible contributors to the development of pancreatic cancer-induced skeletal muscle wasting.

## 1. Introduction

Innate [1] and adaptive immune cells, as well as humoral immune factors [2], can aid in the host-mediated destruction of cancer cells. Cytotoxic T cells (CD8+) are among the most well-defined anti-cancer effectors. For many solid tumors, a high degree of tumor T cell infiltration indicates good prognosis [3,4,5,6,7]. Accordingly, immunotherapies are intended to enhance, direct, or facilitate the natural ability of T cells to destroy cancer cells [8] and have been among the most effective cancer therapies to date. Chimeric antigen receptor T-cell therapy has produced up to a 90% complete remission response in patients with B-cell lymphomas [9], and T-cell checkpoint inhibitors have produced remarkable and durable tumor regression in many different cancer types [10]. However, cancer often employs elusive strategies to evade T-cell immunity and thereby limit the effectiveness of host immunity and immunotherapies alike [8,11]. While immune cells can be protective against cancer, they are not necessarily so. Some immune cells can exert immunosuppressive functions and thereby impair anti-cancer immunity and encourage cancer disease progression.

Secreted factors from the tumor [12] and host cells [13] can initiate an altered state of myelopoiesis, whereby immature myeloid cells (IMC) are preferentially generated from common hematopoietic progenitor cells in the bone marrow at the expense of lymphoid and erythroid cell production [14,15,16]. Many IMCs do not fully mature and develop an immunosuppressive phenotype [16]. This heterogenous class of immunosuppressive cells are called myeloid-derived suppressor cells (MDSCs). Broadly, there are at least two subclasses of MDSCs based on their immunophenotype: those that resemble monocytes (mMDSCs), and those that resemble neutrophils, and are thus called granulocytic MDSCs (gMDSCs). While both mMDSCs and gMDSCs are elevated in tumor-bearing hosts, gMDSCs are expanded to a greater degree in tumor-bearing mice [17] and human cancer patients [18].

MDSCs accumulate in tumors and protect cancer cells from T-cell-mediated destruction. This can be orchestrated by the cancer cells themselves. In pancreatic cancer, oncogenic KRAS mutation drives the expression of chemokines and cytokines that can promote the accumulation of MDSCs to defend primary and secondary tumors from host immunity [19,20,21]. The prognostic significance of MDSCs in patients with solid tumors is grim, with high levels of circulating MDSCs independently predictive of poor overall survival, progression-free survival, and disease-free survival [22]. As such, MDSCs have emerged as immunotherapeutic targets for cancer treatment [14,23].

Neutrophils, which are a distinct cell type from gMDSCs and are not necessarily immunosuppressive, are also expanded in many tumor-bearing hosts and can contribute to cancer disease progression [24]. The absolute number of circulating neutrophils and the neutrophil-to-lymphocyte ratio (NLR) are elevated in patients with multiple types of solid tumors and many studies have shown that elevated neutrophils and a high NLR is associated with poor outcomes [25,26,27].

While the role of MDSCs and neutrophils in cancer disease progression and treatment outcomes is more well understood, whether the advent of MDSCs and expansion of neutrophils over the course of cancer progression contributes to other undesirable impacts on the host—such as cancer cachexia—is less well understood. Cancer cachexia is a catabolic condition, characterized by progressive body weight and skeletal muscle loss with or without loss of fat [28], that affects up to 80% of cancer patients [29]. The presence of cachexia significantly impairs quality of life by promoting a decline in physical function, as well as length of life—with cachexia estimated to be the direct cause of 20 to 30% of all cancer-related mortality [30,31]. Moreover, the presence of cachexia reduces patient candidacy for potentially curative interventions [29,30]. Therefore, strategies that prevent, slow, or reverse cancer cachexia may be tantamount to improving patient survival. While effective anti-cachexia interventions are under development and refinement [32], there remains a need to uncover the basic molecular and cellular mechanisms of cancer cachexia. While cancer cachexia is a complex syndrome impacting multiple tissue types, in this study, we tested the hypothesis that depleting Ly6G+ cells, which include gMDSCs and neutrophils, would reduce muscle atrophy in tumor-bearing mice.

## 2. Materials and Methods

### 2.1. Mice

All procedures involving mice were approved by the University of Florida Institutional Animal Care and Use Committee and were in incompliance with NIH Guidelines for Use and Care of Laboratory Animals. Ten-week-old male C57BL/6J mice, obtained from The Jackson Laboratory (Bar Harbor, ME, USA), were housed in a temperature- and humidity-controlled facility with a standard 12 h light/dark cycle. Standard chow and water were available ad libitum. After arrival at the vivarium, mice were left unperturbed for 1 week prior to starting experiments.

### 2.2. Pancreatic Tumor Model

KPC (*LSL-Kras^G12D/+^; LSL-Trp53^R172H/+^; pdx-1-Cre*) 1245 pancreatic cancer cells (gifted by Dr. David Tuveson (Cold Spring Harbor Laboratory, Cold Spring Harbor, New York, NY, USA)) were cultured in DMEM supplemented with 10% FBS, 1% penicillin, and 1% streptomycin at 37 °C with 5% ambient CO_2_. Medium was refreshed every 48 h. Cells were collected once 80 to 90% confluence was reached. Cells were washed, detached from the cell culture flask using trypsin, and centrifuged at 300× *g* for 5 min. The cell pellet was re-suspended in sterile PBS at 5 × 10^6^ cells per mL and placed on ice. Eleven-week-old C57BL/6 J mice were anesthetized using isoflurane vaporized in pure oxygen. After sufficient depth of anesthesia was attained, the pancreas was exposed and 50 μL of the KPC 1245 cell suspension (2.5 × 10^5^ total cells) was injected into the tail of the pancreas. After the injection, the incisions in the muscle and skin layers were closed using absorbable suture and surgical clips, respectively. The mouse was returned to a home cage where it was monitored until recovery. A subset of mice underwent a sham procedure in which the pancreatic injection delivered PBS alone. All surgical procedures were performed using aseptic techniques. Mice received buprenorphine as a preemptive analgesic and every 12 h for 48 h after the procedure. Tumor-bearing and sham control mice were euthanized 15 days after KPC cell injection or sham procedures. Fifteen days was chosen because we have previously shown [33] that these mice develop cachexia and reach IACUC-mandated tumor endpoint by showing signs of pain and distress by this timepoint.

### 2.3. Ly6G Antibody-Mediated Cell Depletion

It has been shown that 1A8 is a Ly6G-specific monoclonal antibody produced in the rat that is well tolerated and routinely used to deplete Ly6G+ cells in mice [20,34,35]. In this study, one group of tumor-bearing mice received 10 mg/kg of 1A8 antibody (Ly6G) by IP injection. A second group of tumor-bearing mice were treated with an equal amount of rat IgG2a isotype control antibody 2A3 (Isotype). Non-tumor-bearing sham control mice also received either the isotype control antibody or Ly6G antibody. Because no significant treatment effects were seen in the control (non-tumor-bearing) groups, the data from these mice were collapsed into a single group, which is hereafter called the control group. Antibody injections were undertaken at 5, 8, 11, and 14 days after cell injection or sham procedures. Antibodies (1A8 and 2A3) were purchased from BioXcell (West Lebanon, NH, USA). A diagram illustrating the study design is presented in Figure 1.

### 2.4. Tissue Specimen Collection and Storage

Prior to euthanasia, mice were deeply anesthetized with vaporized isoflurane (2 to 3% in pure oxygen). Once sufficiently anesthetized, the *tibialis anterior* (TA) muscles, *gastrocnemius/plantaris* muscle complexes, and epididymal fat pads and tumors were removed and weighed. The diaphragm muscle was also carefully excised. Muscles and tumors were either flash frozen in liquid nitrogen or embedded in optimal cutting temperature (OCT) compound and frozen in liquid 2-methylbutane cooled in liquid nitrogen. Specimens were stored at −80 °C.

### 2.5. Histology

Muscle and tumor samples were cryosectioned (10 μm) and mounted on glass microscope slides. Muscles were cut by cross-section and a complete face of the tumor was sectioned through the middle. The slides were air-dried at room temperature for 1 to 2 h, then stored at −80 °C for later use.

TA muscle cross-sections were stained with Alexa Fluor 594-conjugated wheat germ agglutinin (WGA, Invitrogen, Carlsbad, CA, USA) to visualize muscle fiber borders. To do this, sections were removed from the freezer and air-dried for 1 h. WGA, diluted 1:200 in PBS-Tween 20 (0.1% (*v*/*v*)), was applied to the sections for 1 h in a dark humidified chamber at room temperature. Thereafter, sections were washed 3 times for 5 min in PBS-Tween 20. After the last wash, slides were mounted with glass coverslips using ProLong Diamond Mountant (Invitrogen, Carlsbad, CA, USA).

Diaphragm muscle cross-sections were stained with WGA to visualize muscle fiber borders and monoclonal antibodies against myosin heavy chain type I (MHC I, clone BA-D5) and myosin heavy chain type IIA (MHC IIA, clone SC-71) to identify muscle fiber types. Both myosin heavy chain antibodies were purchased from Developmental Studies Hybridoma Bank (Iowa City, IA, USA). Slides were thawed and air-dried for 1 h then permeabilized in PBS containing 0.1% Triton X-100 at room temperature for 10 min. Sections were blocked in PBS containing 5% normal goat serum and 0.1% Triton X-100 for 60 min at room temperature. After blocking, the primary antibodies (diluted 1:10 in blocking solution) were applied to the sections for 90 min at room temperature in a humidified chamber. Sections were washed then incubated in WGA (1:200) and secondary antibodies (Alexa Flour 488-conjugated goat anti mouse IgG2b at 1:300, and Alexa Flour 350-conjugated goat anti mouse IgG1 at 1:200) for 60 min at room temperature in a dark humidified chamber. Slides were then washed and coverslipped with ProLong Diamond Mountant (Invitrogen, Carlsbad, CA, USA).

Tissue sections were stained with anti-Ly6G antibody (1A8, BioXcell, West Lebanon, NH, USA) to quantify the density of Ly6G+ cells (TA and tumor) and positive staining area (diaphragm). Serial sections from each specimen were stained with the isotype-matched control antibody in parallel to ensure specificity of the stain assay. Slides were removed from the freezer to thaw and air-dry for 1 h. Sections were fixed in 10% formalin then washed 3 × 10 min in PBS-Tween. Endogenous peroxidases were quenched using 3% H_2_O_2_ for 15 min, then washed 3 × 5 min. Sections were serum-blocked for 90 min. Blocking reagent was then tipped off the slide and 4 μL/mL of primary antibody (rat anti-mouse Ly6G (1A8) or isotype control (2A3)) was applied overnight at 4 °C in a humidified chamber. The next morning, slides were washed 3 × 5 min in PBS-Tween. An HRP polymer-conjugated goat anti-rat IgG secondary antibody (VC005, R&D systems, Minneapolis, MN, USA) was applied to the sections for 90 min at room temperature in a humidified chamber. Sections were washed 3 × 5 min in PBS-Tween. The chromogenic substrate (DAB) was added to the sections for 45 s. Sections were promptly rinsed in ddH_2_O and washed in ddH_2_O for 5 min. Sections were then counterstained with hematoxylin, dehydrated in ascending ethanol concentrations, cleared in xylenes, and then coverslipped with Permount mounting medium.

### 2.6. Micrograph Acquisition and Analysis

WGA fluorescent images of the TA were acquired using a Leica TCS SP8 confocal microscope (Leica Microsystems, Bannockburn, IL, USA) at 10 times magnification. Images were stitched to create a single image of the entire cross-section. WGA, MHC I, and MHC IIa-stained diaphragm strips were imaged on a Leica DM 5000B microscope. The entire strip was imaged with non-overlapping images at 10 times magnification. Micrograph quantification was conducted using ImageJ 1.54k software (https://imagej.nih.gov/ij/ accessed date on 24 March 2022). For muscle fiber size measurements, the image of the WGA channel was converted to an 8-bit image and a threshold was set. Individual fibers were traced using the wand tool. Some fibers with impaired or discontinuous membrane staining were traced manually using the freehand selection tool. MHC I- and MHC IIa-positive fibers were classified as type I, and type IIa fibers, respectively. Fibers lacking immunoreactivity for both MHC I and MHC IIa were classified as type IIb, IIx fibers. The whole TA muscle cross-section was traced resulting in > 2000 traced fibers per mouse. A minimum of 1000 diaphragm muscle fibers were traced per mouse. Diaphragm fibers were evenly selected across the diaphragm strip. Fibers that were oblong or longitudinally cut were not traced. Minimum Feret’s diameter (MFD) was used as the metric for muscle fiber size.

Ly6G-stained tumor and TA sections were imaged with the 20× objective on a Leica DM500B microscope. For each tumor section, the entire border was imaged with non-overlapping pictures and 3 to 5 random areas within the middle region were taken. For the TA sections, 6 random images were taken with the 20× objective. Ly6G+ cells were counted in each image and the total number of cells per tumor section was normalized to the number of 20× images analyzed.

Images from the diaphragm muscles stained for Ly6G were processed using the color deconvolution feature in ImageJ. The color deconvolution was achieved using “user values” that were determined using positive and negative control images. A threshold limit was set for the DAB image based on positive and negative control images. The muscle was traced to exclude borders and artifacts, then the percent positive muscle area was measured and recorded.

### 2.7. Atrophy-Related Gene Expression

To obtain RNA samples, tibialis anterior and diaphragm muscles were homogenized in Trizol (ThermoFisher, Waltham, MA, USA). RNA was extracted using chloroform, precipitated using isopropanol, washed in 70% EtOH, and dissolved in nuclease-free water. RNA concentration and integrity were assessed using a Bioanalyzer 2100 (Agilent, Santa Clara, CA, USA). Then, 1μg of RNA was reverse transcribed using a SuperScript IV Reverse Transcriptase kit (ThermoFisher, Waltham, MA, USA). Quantitative fluorometric PCR was performed on the generated cDNA using the following Taqman Probes (ThermoFisher, Waltham, MA, USA): GAPDH (Mm99999915_g1); Atrogin-1 (Mm00499523_m1); and MuRF-1 (Mm01185221_m1). Gene expression data were calculated using the ΔΔCt method with GAPDH as the reference housekeeping gene.

### 2.8. Statistics and Data Presentation

Muscle mass, fiber size, and gene expression variables were analyzed using a one-way ANOVA with group (Vehicle, Isotype, Ly6G) as the independent variable. When a significant main effect was found, pairwise comparisons were made using a Tukey’s HSD test. Tumor weight and histology variables were analyzed with independent *t*-tests. Comparisons were considered statistically significant when *p* < 0.05. Statistical analyses were conducted using Graphpad Prism version 8.3.1 (San Diego, CA, USA). All data are provided as mean ± SD. Graphical figures were made using Graphpad Prism software.

## 3. Results

### 3.1. Anti-Ly6G Treatment Delays Body Wasting and Reduces Loss of Skeletal Muscle Mass

KPC tumor-bearing Isotype-treated mice began losing body weight on day 8, which continued to decrease until study endpoint, whereas KPC anti-Ly6G-treated mice did not begin losing body weight until day 12, which then also progressively continued until endpoint (Figure 2A). As expected, at study endpoint (day 15), Isotype-treated KPC mice showed significant loss of skeletal muscle mass in both the *tibialis anterior* (TA) muscle (control, 51.2 ± 5.2 g; KPC Isotype 37.1 ± 3.4 g; Figure 2B) and the *gastrocnemius/plantaris* muscle complex (control, 162.7 ± 16.9 g; KPC Isotype, 125.4 ± 8 g; Figure 2C). However, in KPC anti-Ly6G-treated mice, this loss of TA muscle mass was significantly attenuated (44.9 ± 3.4 g), and the GP mass did not significantly atrophy (139.7 ± 15.9 g) compared to control mice. Both KPC tumor-bearing groups had significant epididymal fat wasting compared to sham control mice; no significant differences in endpoint fat mass were observed between Isotype- and anti-Ly6G-treated tumor-bearing mice (Figure 2D). Combined, these data show that anti-Ly6G treatment attenuates skeletal muscle atrophy and delays body weight loss in KPC tumor-bearing mice.

### 3.2. Anti-Ly6G Treatment Depletes Ly6G+ Cells in the Tumor and TA Muscle

Anti-Ly6G treatment did not have a significant impact on tumor growth, with tumor mass similar between isotype control and anti-Ly6G-treated mice at study endpoint (Figure 3A). Staining of tumors with anti-Ly6G antibody revealed Ly6G+ cells with the characteristic polymorphonuclear shape of granulocytes (Figure 3B) present within KPC tumors of both isotype control- and anti-Ly6G-treated groups. However, tumors from anti-Ly6G-treated mice had significantly fewer Ly6G+ cells per field of view (FOV) (isotype; 19.9 ± 3.9 vs. Ly6G; 6.9 ± 3.9, *p* = 0.0015, Figure 3C,D) producing a ~65% depletion efficiency in the tumor. Specificity of the Ly6G stain was further demonstrated via staining of tumors with isotype-matched control antibody in lieu of the anti-Ly6G antibody, which revealed no reactivity in tumors of isotype-treated mice. However, in tumors from anti-Ly6G-treated mice, immunoreactivity of some cells to the anti-rat secondary antibody was revealed (Figure 3C), likely reflecting Ly6G+ cells bound, but not yet depleted, by the rat anti-Ly6G antibody during treatment.

KPC tumor-bearing mice also showed a significant increase in the number of Ly6G+ cells in the TA muscle, an effect that was significantly attenuated by anti-Ly6G treatment (Figure 4A,B), with a depletion efficiency (~50%) similar to that observed in the tumor. The area of Ly6G+ staining in the diaphragm was likewise significantly greater in tumor-bearing animals; however, the attenuation by anti-Ly6G treatment did not reach statistical significance in this tissue (*p* = 0.18) (Figure 4A,B).

### 3.3. Anti-Ly6G Treatment Attenuates Muscle Fiber Atrophy

To further determine whether the protection against KPC-induced muscle atrophy provided by anti-Ly6G treatment carries over to a protection against fiber atrophy, we measured TA muscle fiber size by measuring the minimum Feret’s diameter (MFD). As expected, TA muscle fiber size was significantly smaller in Isotype-treated KPC mice compared to control mice (MFD: 37.8 ± 1.7μm vs. 41.3 ± 1.5 μm, *p* = 0.004). However, TA fiber size of KPC mice treated with anti-Ly6G antibody was significantly larger (MFD: 40.7 ± 1.3 μm) than Isotype-treated KPC mice (*p* = 0.03) and not different to control mice (*p* = 0.85; Figure 5A,B). These findings indicate that anti-Ly6G treatment offered near complete protection against tumor-induced TA muscle fiber atrophy.

We also measured muscle fiber size in the diaphragm, since this muscle similarly undergoes significant wasting in tumor-bearing hosts [36]. Diaphragm muscle fiber size was significantly smaller in both Isotype-treated (MFD: 19.1 ± 1.4 μm) and anti-Ly6G-treated (MFD: 21.2 ± 1.9 μm) KPC groups compared to control mice (MFD: 26.4 ± 4.4 μm, Figure 6B). Although diaphragm fiber size was numerically larger in anti-Ly6G-treated compared to Isotype-treated KPC mice, this difference did not reach statistical significance (*p* = 0.08, Figure 6B). However, because the diaphragm is heterogeneous in its fiber type distribution, we also considered the potential impact of anti-Ly6G+ treatment on fiber type-specific atrophy. Interestingly, although we found a similar degree of atrophy in type I and type IIa diaphragm fibers in both KPC tumor-bearing groups, type IIx/IIb fibers showed significantly less atrophy in KPC mice treated with anti-Ly6G antibody (*p* = 0.03, Figure 6B). This indicates that anti-Ly6G treatment in KPC mice was protective against diaphragm muscle fiber atrophy specifically in IIx and IIb fibers.

### 3.4. KPC Tumor Burden Causes a IIx/b to IIa Fiber Type Shift Independent of Ly6G Treatment

Since we fiber typed the diaphragm to evaluate fiber type-specific atrophy, we subsequently considered whether anti-Ly6G treatment impacts muscle fiber type distribution in KPC tumor-bearing mice. In control mice, the most abundant fiber type observed was IIb/IIx, which made up 56.9 ± 2.7% of all fiber types (Figure 6C). This percentage distribution significantly decreased in both KPC tumor-bearing groups (KPC Isotype; 50.9 ± 3.5%, KPC Ly6G; 51.2 ± 0.8%, both *p* values < 0.009, Figure 6C), with no difference between Isotype and anti-Ly6G groups (*p* = 0.99). Type IIa fibers made up 34.9 ± 3.0% of all diaphragm fibers in control mice, and KPC tumor burden led to a significant increase in this percentage to a similar degree in both treatment groups (KPC Isotype; 40.3 ± 3.4%, KPC anti-Ly6G; 40.1 ± 0.6%, both *p* < 0.02 Figure 6C). There was no significant difference in IIa fiber frequency between KPC Isotype-treated and KPC anti-Ly6G-treated mice (*p* = 0.99). Type I fibers made up 7.9 ± 0.9% of all diaphragm fibers in control mice, and KPC tumors had no significant impact on this distribution (KPC Isotype 8.7 ± 0.9%, *p* = 0.3; KPC Ly6G (8.6 ± 0.9%, *p* = 0.3 Figure 6C). These data indicate that KPC tumors cause a fiber type shift away from IIx/IIb fibers toward IIa fibers and that the anti-Ly6G treatment had no impact on this shift.

### 3.5. Anti-Ly6G Treatment Attenuates Atrophic Gene Expression in TA Muscle

Based on our finding that KPC mice treated with anti-Ly6G antibody showed preservation of both muscle mass and muscle fiber size in the TA muscle, we further assessed the mRNA levels of the muscle atrophy-associated E3 ligases, Atrogin-1/*Fbxo32* and MuRF-1/*Trim63*. Compared to control mice, atrogin-1/*Fbxo32* and MuRF-1/*Trim63* were profoundly increased in the TA muscles of KPC mice treated with Isotype control antibody, which were significantly attenuated in KPC mice treated with anti-Ly6G antibody (Figure 7). In contrast, anti-Ly6G treatment did not attenuate the upregulation of either gene in the diaphragm in response to KPC tumor burden (Figure 7).

## 4. Discussion

Altered myelopoiesis resulting in the expansion of neutrophils and MDSCs (particularly gMDSCs) occurs with many cancers [16,18]. It is well known that the advent of these cells poses a threat to effectiveness of immunotherapy and anti-cancer immunity. Accordingly, in patients with several types of solid tumors, circulating MDSC frequency is positively related to cancer stage and is negatively correlated with time to progression [16]. In addition to their established immunosuppressive and pro-tumorigenic effects, a potential role of MDSCs in cancer cachexia has been suggested because: (1) they can secrete cachectogenic cytokines [15,37,38]; (2) there is a temporal association between MDSC expansion and cachexia [39]; and (3) MDSCs can alter amino acid availability [15,37,40], which may deprive the muscle of anabolic signals and substrates.

This investigation is among the first to directly test the hypothesis that gMDSCs and/or neutrophils indeed contribute to pancreatic cancer-induced muscle atrophy. Here, we used an antibody-mediated approach to deplete cells expressing the Ly6G antigen. Ly6G belongs to a large family of Ly6 proteins that are expressed on the cell surface of leukocytes in a lineage-specific fashion. As such, the expression of Ly6G is highly restricted to gMDSCs and neutrophils [41]. We show that anti-Ly6G treatment of KPC mice reduced the density of Ly6G+ cells in skeletal muscles and tumors. The treatment also reduced important features of cancer cachexia, including body weight loss, loss of skeletal muscle mass, and muscle fiber atrophy in a limb (TA) and respiratory (diaphragm) muscle. Moreover, we further show that the increased level of the atrophy-related genes, *Atrogin-1* and *MuRF1*, in the TA of KPC mice is reduced in the Ly6G-treated KPC mice.

As mentioned above, other investigators have established associations between MDSCs and cachexia, and have discussed sound theoretical contributions of MDSCs to cachexia. To our knowledge, however, few studies have directly tested the hypothesis that MDSCs contribute to cancer cachexia. In one such study [42], mice bearing tumors produced from the 4T1 mammary carcinoma cell line displayed MDSC expansion and developed cachexia. However, mice in the same study bearing tumors produced from the 66C4 cell line (a subclone of the 4T1 cell line) did not display MDSC expansion and did not develop features of the cachexia phenotype—i.e., hepatic acute-phase protein response, hypermetabolism, and fat wasting. Because the cell lines were closely related (parent cell line vs. sub clone), the finding that 4T1 cells produced MDSC expansion and cachexia while the 66C4 cells produced neither is consistent with the hypothesis that MDSCs contribute to the development of cancer cachexia. However, this experimental design is open to an important confounding factor: closely related cancer cells can produce substantially different immunomodulatory effects in the host [7]. Therefore, the MDSC expansion in the 4T1 tumor-bearing mice could have been a covariate with cachexia rather than a cause of cachexia. Our results align with this study and build on these findings because the anti-Ly6G treatment used here reduced muscle loss compared to mice bearing tumors from the same cell line, indicating that gMDSCs and/or neutrophils may indeed contribute to pancreatic cancer-associated muscle loss. Moreover, in the study by Cuenca et al. [42], the researchers focused on other features of cachexia (metabolic responses, fat mass, hepatic acute-phase protein response). The present study provides novel data showing that anti-Ly6G treatment attenuated hallmarks of cachexia (i.e., the loss of body mass and skeletal muscle) in this orthotopic KPC model.

Because Ly6G is not restricted to gMDSCs only, but is also present on neutrophils [41], we cannot rule out the involvement of neutrophils, which are also depleted by the anti-Ly6G treatment. In fact, it could be the sheer expansion of granulocytic immune cells, rather than a specific phenotype of these cells, that contributes to cachexia. In support of this, adoptive transfer of large numbers of CD11b+ and GR1+ leukocytes from tumor-bearing mice (which would include inflammatory monocytes, neutrophils, and gMDSCs [41]) or CD11b+ and GR1+ cells from healthy mice (which would confer inflammatory monocytes and neutrophils, but few gMDSCs) similarly activated an acute-phase protein response [42]. While that experiment did not look at skeletal muscle wasting, it did show that increasing the number of circulating CD11b+ and GR1+ cells, whether from tumor-bearing mice or heathy mice, generated an acute-phase response, which is associated with cachexia [43]. A future experiment assessing markers of skeletal muscle wasting after performing adoptive cell transfer of specific cell types (e.g., gMDSCs or neutrophils) would be valuable to follow up this research.

Other evidence that sheer neutrophil expansion may contribute to cachexia includes the relationship between neutrophil-to-lymphocyte ratio (NLR) and cachexia. NLR is a common blood test that displays the dynamic relationship between innate and adaptive cellular branches of the immune system. Elevated NLR ratios are commonly observed in cancer patients and are used as an index of systemic inflammation. Elevated NLR also likely indicates altered hemopoiesis associated with cancer, which is skewed in favor of myeloid cell production [16]. Multiple studies have shown that elevated NLR is positively associated with cancer cachexia [25,26,27,44]. One such study [26] conducted gene transcriptomic analysis in blood cells of patients deemed to have no cachexia, pre-cachexia, or cachexia. The bioinformatics analysis of these data predicted that the expression of the neutrophil-derived proteases cathepsin B and G were master regulators of multiple genes believed to be pro-cachectic, including interleukin 8 (IL-8) and transforming growth factor beta (TGF-β) members. The authors concluded that “unbridled neutrophil activity” may cause the development and progression of cachexia. Our results are in agreement with this suggestion since the anti-Ly6G treatment used in our study would also deplete neutrophils.

MDSCs, particularly the gMDSC subset, are elevated in tumor-bearing mice and human cancer patients [16], and accumulate in the tumors and circulation. As expected, based on this previous literature, we show that Ly6G+ cells were present in the tumors and that anti-Ly6G treatment reduced the density of these cells in the tumor by more than 50%. A novel finding of this study is that Ly6G+ cells were also elevated in the limb and diaphragm skeletal muscles of tumor-bearing mice and that anti-Ly6G treatment attenuated this effect to a similar magnitude as in the tumors. The Ly6G+ cells present in skeletal muscles of tumor-bearing mice were localized to the extracellular microenvironment, and had not infiltrated muscle fibers themselves, which is a histological feature of injured skeletal muscle. While several immune cells, including monocytes, macrophages, and T-cells, have been found to be increased in the skeletal muscle tissue of tumor-bearing hosts [45,46], this is the first report, to our knowledge, that Ly6G+ cells have been observed.

While the present study supports the conclusion that gMDSCs and/or neutrophils contribute to pancreatic cancer-induced muscle atrophy, this study was not designed to test the mechanisms by which Ly6G+ cells do so. Though, this now warrants further investigation. Because elevated numbers of Ly6G+ cells were observed in atrophied skeletal muscles of tumor-bearing mice, which were reduced concomitantly with the sparing of muscle mass in tumor-bearing mice treated with anti-Ly6G antibody, one possible mechanism is a direct and local effect of Ly6G+ cells on the skeletal muscle microenvironment. Indeed, MDSCs are known to secrete several cachectogenic factors including tumor necrosis factor alpha (TNF-α), TGF-β members, nitric oxide, and peroxynitrate [47]. These factors can either act directly on muscle cells to increase intracellular catabolic signaling pathways that induce muscle atrophy, as is the case for TNF-α [48] and TGF-β members [49] or, in the case of myeloid-derived nitric oxide and peroxynitrite, can contribute to skeletal muscle damage and degeneration [50], which are pathologic features and potential contributing factors of cancer cachexia [46]. Anti-Ly6G treatment alone did not reduce tumor burden. This is not surprising since previous investigations have shown that Ly6G+ cell depletion through treatment with 1A8 antibody enhances the effectiveness of concurrent immunotherapy but is not consistently effective in decreasing tumor size or survival when used in isolation [20,51,52]. The fact that tumor burden was not different between treatment groups indicates that the cachexia-mitigating effect of the anti-Ly6G treatment is not due to slowing tumor progression. Instead, the reduced numbers of Ly6G positive cells within the tumor microenvironment resulting from anti-Ly6G treatment could impact the cachectogenic potential of the tumor. In this regard, a variety of PDAC tumor changes have been documented in response to 1A8 antibody-mediated Ly6G+ cell depletion including cell population infiltrate, blood vessel diameter, and extra-cellular matrix remodeling [23], thus lending credence to a possible indirect effect of gMDSCs on muscle atrophy, such that their presence increases the cachectogenic potential of the tumor, rather than the overt mass of the tumor. Moreover, gMDSCs are instrumental in promoting metastatic disease, and gMDSC depletion via anti-Ly6G treatment has been shown to reduce metastatic disease in preclinical models [20,53]. Since metastatic cancer is positively associated with cachexia [54], it is possible that the anti-Ly6G treatment reduced cachexia secondary to reducing metastasis. Unfortunately, we did not measure metastasis in this study.

Depletion of Ly6G+ cells in tissues aside from skeletal muscle and tumor—such as the brain—could also impact cachexia. Indeed, work by Burfeind et al. showed that cells expressing CD45+ CD11b+ Ly6G+ surface markers (consistent with both neutrophils and gMDSCs) accumulate in the brain of KPC mice and contribute to anorexia and skeletal muscle atrophy [55]. Interestingly, the authors discovered that these cells were recruited to the brain via a CCR2-dependent signaling axis. Based on this finding, and the failure of the 1A8 antibody to successfully deplete Ly6G+ cells in their study, the authors subsequently reduced the recruitment of these cells to the brain by disrupting the CCR2 chemotactic axis. While our data are consistent with these findings, in that Ly6G+ cells are implicated in pancreatic cancer-associated muscle atrophy, the methods used to deplete the Ly6G+ cells are different and there is discrepancy between studies in whether 1A8 can successfully deplete Ly6G+ cells in the KPC model. Certainly antibody-mediated depletion of neutrophils and gMDSCs is routinely undertaken using 1A8 [56], which binds to Ly6G in a highly specific manner. In our study, we have confidence that the 1A8 antibody did successfully deplete Ly6G+ cells, with an efficiency of at least 50%, based on their measurement in tumor and skeletal muscle using immunohistochemistry. This outcome measure is not subject to the same competitive inhibition (antigen masking) that was observed by Burfeind et al. [55] when using flow cytometry. Nonetheless, since we did observe immunoreactivity of some cells with the (anti-rat) secondary antibody within tumors of mice treated with the rat anti-Ly6G antibody, this indicates that at least some Ly6G cells were bound by antibody, but not yet depleted.

One possible reason for the discrepancy in the 1A8-mediated depletion efficiency between our study and that of Burfeind et al. may be related to differences in the treatment protocol. Our method was more acute and less frequent, providing four total injections, at 5, 8, 11, and 14 days after tumor cell injection. In contrast, Burfeind et al. began antibody treatment earlier (2 days post KPC cell injection) and administered treatment more frequently (daily). It is possible that this frequency of treatment could have exaggerated a “rebound” effect resulting in accelerated Ly6G+ cell production in response to depletion pressure [56].

An interesting finding of this study is an apparent fiber type-specific protective effect of the anti-Ly6G treatment. Myofiber atrophy in cancer cachexia is reportedly preferential to the fast twitch muscle fibers [57]. The data presented here are consistent with this: the type IIb and IIx fibers showed the largest difference between tumor-bearing mice and the controls, and anti-Ly6G treatment was effective at attenuating this feature of cancer cachexia by inhibiting IIb/IIx fiber atrophy in the diaphragm. This protection against type IIb/x fiber atrophy in the diaphragm without attenuating the increased levels of Atrogin-1 and MuRF1 suggests other atrophy markers or pathways not measured could be impacted by anti-Ly6G treatment. Moreover, because the anti-Ly6G treatment offered a protective effect to the fast fiber types preferentially, this may explain why anti-Ly6G treatment produced a more pronounced preservation to the TA muscle, which is almost purely fast twitch [58].

## 5. Conclusions

In conclusion, this study demonstrates that the depletion of Ly6G+ cells attenuates skeletal muscle atrophy in mice bearing orthotopic KPC pancreatic tumors. This study shows that in addition to the well-known immunosuppressive and/or pro-tumorigenic effects of Ly6G+ cells, they also contribute to cancer cachexia by promoting muscle atrophy. Although the mechanism by which these cells do so is unclear, the presence of Ly6G+ cells in the skeletal muscle of cachectic hosts, and the reduced cachexia following their depletion, suggests there could be a possible direct effect of these cells in the skeletal muscle microenvironment.

## Figures and Tables

**Figure 1 cells-11-01893-f001:**
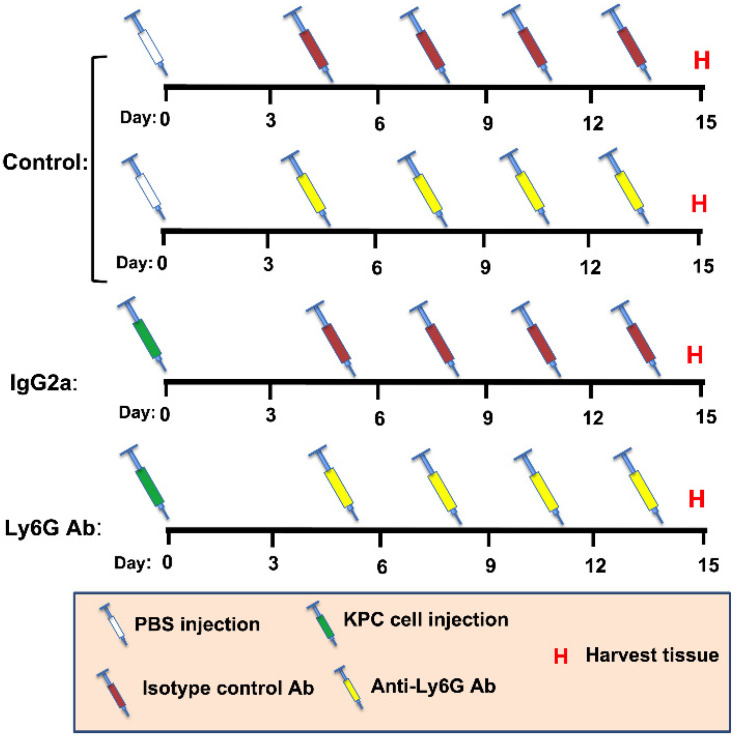
Schematic showing study design. Mice received an orthotopic injection of either PBS (Sham, *n* = 8) or KPC cells (KPC, *n* = 9)) into the pancreases (Day 0). Sham mice and KPC mice were then split into either an isotype control (IgG2a) group or an anti-Ly6G Ab treatment group, to deplete Ly6G-expressing cells, including granulocytic MDSCs and neutrophils. Mice were injected with either IgG2a or Ly6G Ab on day 5, 8, 11, and 14, followed by tissue harvest (H) on day 15. Based on comparable changes in body mass and muscle mass throughout study duration, Sham mice receiving IgG2a and Ly6G Ab were subsequently combined into a single control group (Control).

**Figure 2 cells-11-01893-f002:**
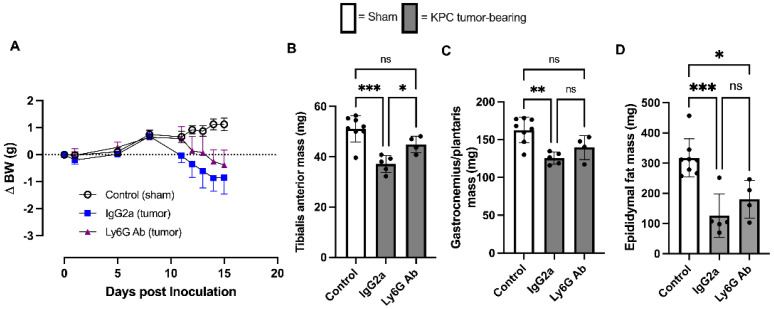
Effect of anti-Ly6G antibody on body weight and skeletal muscle mass in the orthotopic KPC model of cancer cachexia. (**A**) Changes in body weight (mean and standard deviation) of Sham mice (Control; *n* = 8) and KPC tumor-bearing groups treated with either isotype control (IgG2a; *n* = 5) or anti-Ly6G antibody (*n* = 5) throughout study duration. (**B**) Tibialis anterior (**C**) and gastrocnemius/plantaris and (**D**) epididymal fat pad mass from control mice and KPC mice treated with isotype control or anti-Ly6G antibody at experimental endpoint. Open bars = control group; grey bars = tumor bearing. * *p* < 0.05; ** *p* < 0.01; *** *p* < 0.001; ns = not significant (*p* > 0.05).

**Figure 3 cells-11-01893-f003:**
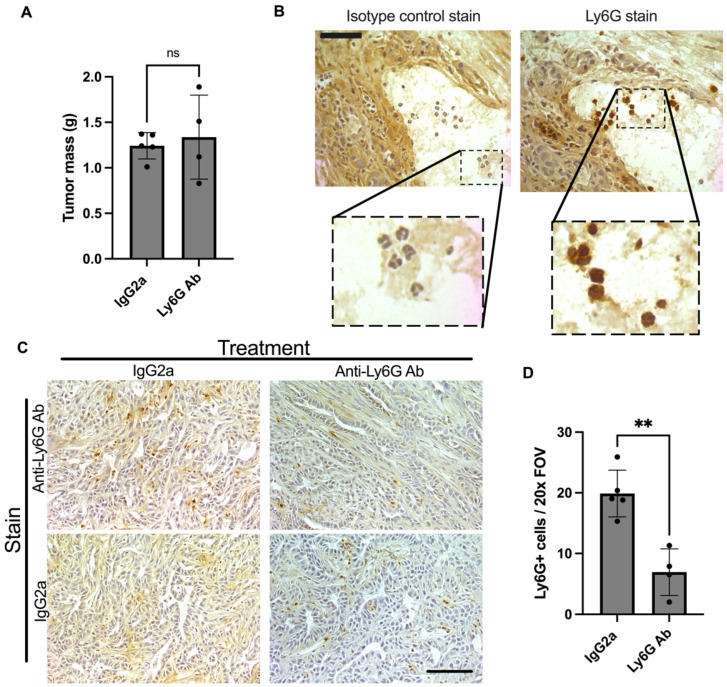
Effect of Ly6G antibody treatment on tumor. Orthotopic KPC tumor-bearing mice were treated with an anti-Ly6G antibody (1A8) or isotype control (2A3) every 3 days starting at 5 days post inoculation. (**A**) Endpoint tumor mass was not different between groups (ns = not significant). (**B**) Serial sections were also stained using the isotype control antibody (2A3) in parallel. This shows a high magnification (40×) view of serial sections. Comparing the stains shows that cells with characteristic multilobed nuclei pick up the Ly6G stain. Scale bar = 50 μm. (**C**) Representative tumor sections stained with anti-Ly6G antibody (1A8) to detect Ly6G+ cells in the tumor. While KPC tumors from isotype control-treated mice did not show positive immunoreactivity in the absence of the Ly6G antibody, KPC tumors from mice treated with anti-Ly6G treatment showed some cells with positive immunoreactivity. These cells likely reflect Ly6G+ cells labeled, but not yet depleted, by the 1A8 antibody in vivo (during treatment). Images were taken at 20× magnification. Scale bar = 100 µm. (**D**) KPC tumors from isotype control-treated mice (*n* = 5) had significantly more Ly6G positive cells per field of view (FOV) than those from anti-Ly6G antibody-treated mice (** *p* < 0.001; *n* = 4).

**Figure 4 cells-11-01893-f004:**
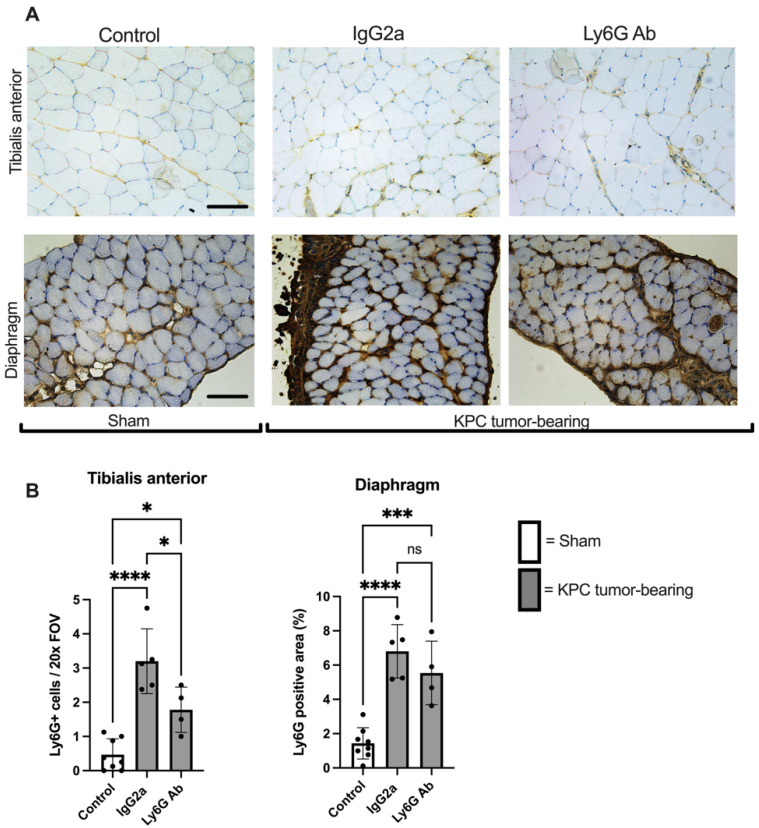
Increased presence of Ly6G+ cells in limb muscles of KPC mice is blunted with anti-Ly6G Ab treatment. (**A**) Representative cross-sectional images of tibialis anterior and diaphragm muscles from Sham (Control; *n* = 8) and KPC mice treated with either isotype control antibody (*n* = 5) or anti-Ly6G antibody (*n* = 4) were stained for the Ly6G antigen. Images were analyzed and are quantified in panel (**B**). Four randomly selected 20× images were analyzed for each mouse of each group. Open bars = Sham (Control). Grey bars = KPC. * *p* < 0.05, *** *p* < 0.001, **** *p* < 0.001, ns = not significant. Scale bars = 100 µm.

**Figure 5 cells-11-01893-f005:**
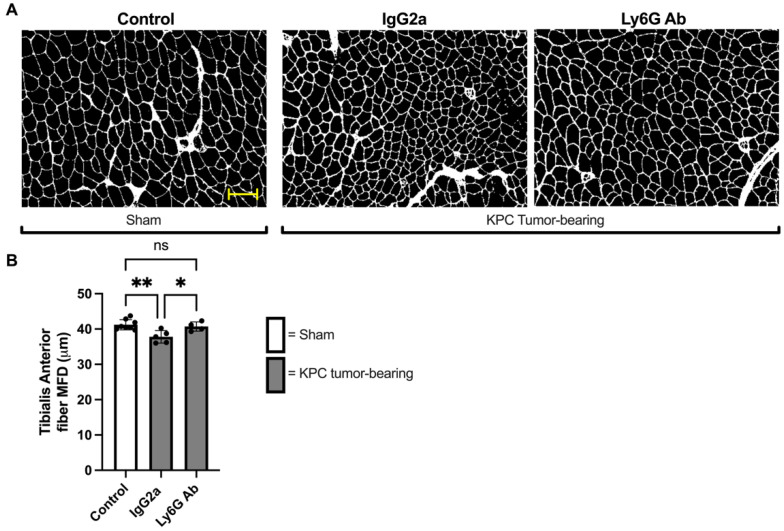
The depletion of Ly6G+ cells protects against orthotopic KPC-induced muscle fiber atrophy. (**A**) Representative images of TA muscle cross-sections from Sham and KPC tumor-bearing mice stained with WGA to outline muscle fiber borders. Scale bar = 100 μm. (**B**) Average TA muscle fiber size, based on quantification of minimum Feret’s diameter (MFD). All intact muscle fibers of the TA muscle were measured. Control: *n* = 8; IgG2a *n* = 5; Ly6G Ab: *n* = 4. * = *p* < 0.05; ** *p* < 0.01; ns = not significant (*p* > 0.05).

**Figure 6 cells-11-01893-f006:**
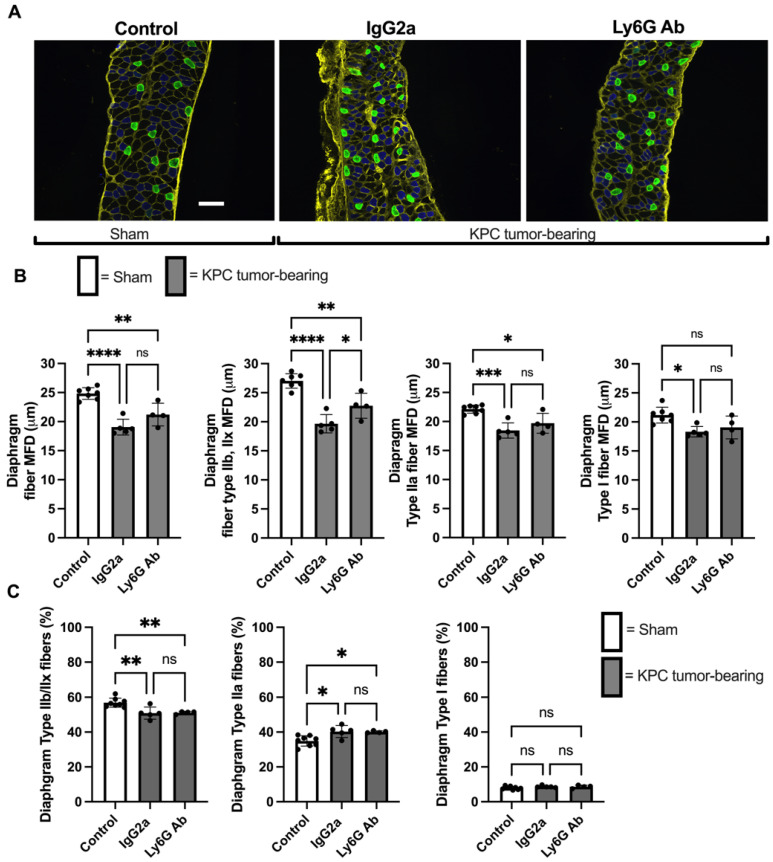
Depletion of Ly6G+ cells protects against Type IIx/b muscle fiber atrophy within diaphragm muscles of KPC mice. (**A**) Representative images of diaphragm strips stained with WGA to label extracellular matrix (yellow), and antibodies against myosin heavy chain I (green) and MHC IIa (blue). Type IIb and IIx fibers were identified as unstained (black) fibers. (**B**) Quantification of average, and fiber type-specific muscle fiber size, based on measurement of the minimum Feret’s diameter (MFD). A minimum of 1000 randomly selected fibers were analyzed per mouse. * *p* < 0.05; ** *p* < 0.01; *** *p* < 0.001; **** *p* < 0.0001; ns = not significant (*p* > 0.05). Control: *n* = 8; IgG2a *n* = 5; Ly6G Ab: *n* = 4. Scale bar = 100 μm. (**C**) Fiber-type proportions in diaphragm muscles from Sham control mice, and KPC tumor-bearing mice. A minimum of 1000 randomly selected fibers were analyzed per mouse. * *p* < 0.05; ** *p* < 0.01; ns = not significant (p > 0.05).

**Figure 7 cells-11-01893-f007:**
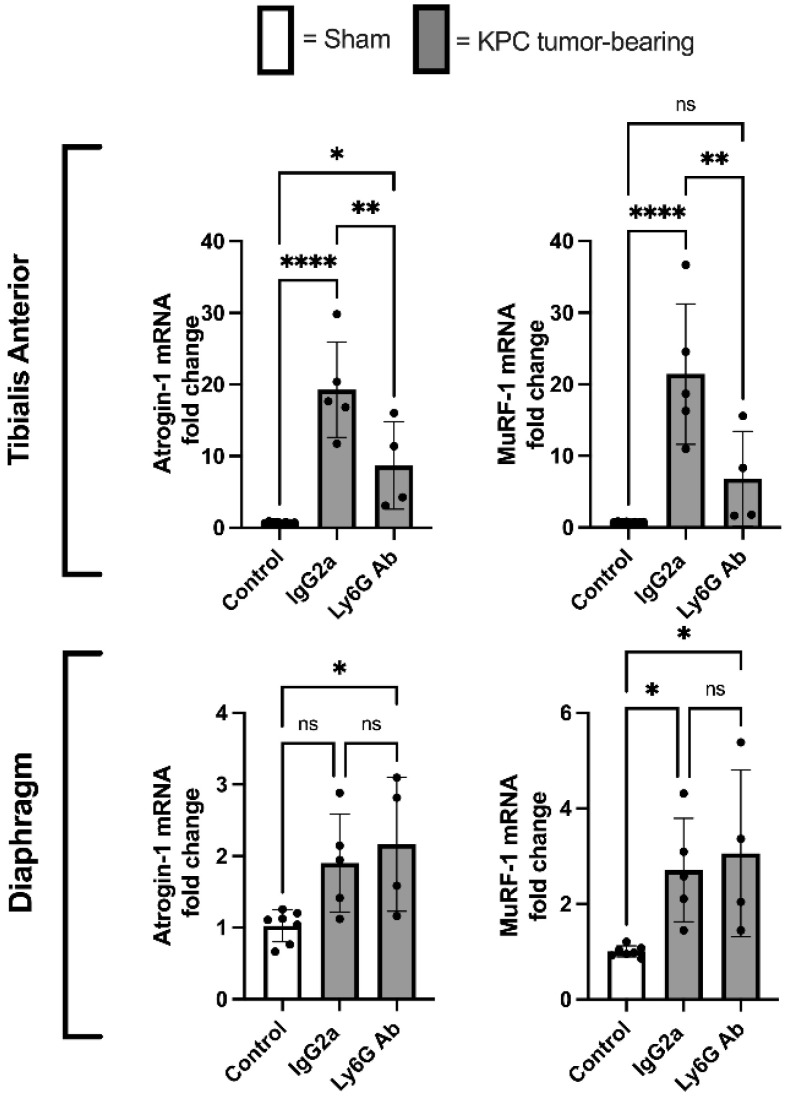
The depletion of Ly6G+ cells inhibits the KPC-induced upregulation of the muscle-specific E3 ligases, *atrogin1/Fbxo32* and *MuRF1/Trim63*. RT-qPCR-based measurement of atrogin-1*/Fbxo32* and MuRF1*/Trim63* mRNA in TA and diaphragm muscles of Sham mice (*n* = 8), and KPC treated with either isotype control (IgG2a; *n* = 5) or anti-Ly6G antibody (*n* = 4). Gene expression data are normalized to GAPDH as a reference gene. PCR reactions were run in triplicate and the average Ct value was used for analysis. * *p* < 0.05; ** *p* < 0.01; **** *p* < 0.0001; ns = not significant (*p* > 0.05).

## Data Availability

The data sets generated from this research may be provided upon reasonable request to the corresponding author.

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
