# Peer review of "Depleting Ly6G Positive Myeloid Cells Reduces Pancreatic Cancer-Induced Skeletal Muscle Atrophy"

_cells, 2022, doi:10.3390/cells11121893_

Round 1

Reviewer 1 Report

 Deyhle et al. present an investigation into the role of Ly6G+ cells in a mouse model of cancer-mediated cachexia. The authors demonstrate increases in Ly6G+ cells in muscle of tumor-bearing mice and that their depletion reduces the number of these cells and moderately preserves muscle mass and fiber size. Despite the fact that this manuscript increases our understanding of the role of immune cells in cancer cachexia, a few minor issues should be addressed prior to publication.

Major concern

Without providing evidence that MDSCs were the major depleted population, the manuscript title should be revised to reflect that all Ly6G cells were depleted. As the abstract states, it is likely that both MDSCs and neutrophils were depleted and either cell type may have been the source of the atrophy reduction.

Minor concerns

While the discussion does an excellent job of balancing the considerations of the potential roles of MDSCs or neutrophils in Ly6G depletion-mediated projection, manuscript’s introduction could be more balanced to acknowledge that neutrophil depletion may also have also played a role in the protection. Many readers may not read the full discussion to appreciate that MDSCs may not be the major reason for Ly6G-mediated protection.

The email address for Andrew D’Lugos is a repeat of Daria Neyroud’s.

Greek characters in the discussion lost formatting in conversion to PDF.

Reviewer 2 Report

Myeloid-derived suppressor cells (MDSCs) are pathologically activated neutrophils and monocytes with potent immunosuppressive activity. While many studies supporting the suppressive role of MDSCs in the regulation of immune responses at different stages of tumor development, the other undesirable impact of MDSCs on host is less well understood. In this study, Michale et al. investigated whether granulocytic MDSCs (gMDSCs) are involved in promoting cancer cachexia. The authors demonstrated a correlation between gMDSCs and cachexia. Depletion of gMDSCs by anti-Ly6G treatment reduced tibialis anterior muscle wasting, abolished TA muscle fiber atrophy, reduced diaphragm muscle fiber atrophy of type IIb and IIx fibers, and reduced atrophic gene expression in the TA muscles. Overall, the manuscript is interesting and well written. However, the strength of the results is limited by the single tumor model used for the study.

Major point:

The authors must add one more tumor model.

Minor point:

The manuscript needs to be carefully checked for typos. (Example, Figure 2 legend, “anit”; Line 423 TGF-?)

Reviewer 3 Report

Manuscript ID: Cells-1745928

Title: “Granulocytic myeloid cell depletion reduces skeletal muscle atrophy in a preclinical model of cancer cachexia’’

In the manuscript, Deyhle et al. investigate the involvement of MDSCs in promoting cancer cachexia in a pancreatic preclinical model. The study's topic is important and valuable as pancreatic adenocarcinoma is a highly aggressive malignancy characterized by rapid progression, exceptional resistance to all forms of anticancer treatment, and a high propensity for metastatic spread. Furthermore, cancer cachexia represents an unmet clinical need. Due to its complexity, insights into novel pathomechanisms and their potential therapeutic applications are direly needed to improve the patient’s clinical situation. In this respect, the current manuscript lacks clinical validation of critical regulators and therefore does not support the conclusions drawn in the text. Additionally, some of the claims are not fully convincing. I have several conceptual and experimental concerns that should be addressed to create a stronger paper that more clearly demonstrates its “innovation” in the field prior to publication. 

Comments:

1.      The author has not provided any data to assess whether only myeloid cells are expanded in pancreatic cancer cachexia in mice, as opposed to other circulating and tissue-resident immune cells reported previously. A relevant experiment should be performed to conclude the findings. 

2.      Recently, an important biological role of MDSCs has emerged that indicates that despite similar origins with many morphological and phenotypic features, both gMDSC and neutrophils have distinct biological roles. How does the author infer their finding from this notion?

3.      Also, based on MDSC phenotypic and morphological features, two major MDSC subsets have been recently defined: polymorphonuclear (PMN) and monocytic (M)-MDSC. Please use this newly defined nomenclature to replace the term granulocytic MDSC and re-edit the title of the manuscript.

4.      Regarding the above comment, PMN-MDSC, but not neutrophils, are immunosuppressive and mouse transcriptomic analysis revealed a clear difference between PMN-MDSC and neutrophils from tumor-bearing mice. Please discuss this rationally.

5.      I have trouble understanding why they focus on gMDSC and neutrophils? Do both infiltrate to the same level in sham and in tumor-bearing animals? How can both contribute to cachexia? Please discuss rationally and provide citations to support their finding. 

6.      The authors highlight the gMDSC and neutrophil population in cancer cachexia. However, the prognostic importance of the NLR (neutrophil-lymphocyte ratio) or other pro-inflammatory measures in the pre-clinical model is not shown? Please provide data. 

7.      What factor from pancreatic cancer causes neutrophils to infiltrate? An answer to this question would add a great deal of interest to the work. 

8.      Several studies in pancreatic cancer have clearly indicated that neutrophil depletion inhibits tumor growth and metastasis (PMID: 24555999; 27265504; 32860704). How does the author infer their finding of no significant impact on tumor growth between isotype and anti-Ly6G treated mice with these studies? Please justify rationally. 

9.      Although initially useful to deplete Ly6G+ neutrophils, the use of anti-Ly6G clone 1A8 suffers some limitations as it is a rat IgG2a that induces Fc-dependent opsonization and phagocytosis of targeted cells and has been associated with a lower efficiency resulting in contradictory experimental results in recent studies. Moreover, it has been recently shown that surface staining for neutrophil markers faces the problem of antigen masking by the used depletion antibody, and thus is not appropriate to assess successful depletion. Please show blood data in this regard to correlate the findings. 

10.  The phenotypic characterization of the animal studies is premature. It is particularly surprising that the manuscript exclusively focuses on skeletal muscle as an indicator for cachexia. In fact, many studies have shown that adipose tissue loss is also a key component of this disease, and certain changes in food consumption are expected to be reflected by alterations in body weight and body fat content. Consequently, the authors need to provide a more detailed phenotypic characterization of their animal models by providing data on body weight, body fat content, fat depot size and histology, energy expenditure, and fecal energy excretion to have a complete view of energy homeostasis. 

11.  How do you explain any (even if minimal) effect on muscle mass without alterations in proteasomal markers etc? 

12.  Does gMDSC or neutrophils from tumor-bearing mice induce cachexia in healthy mice? The authors need to clarify this to establish a convincing functional role of MDSC in mediating cachexia.

13.  Information about how many replicates per experiment were analyzed is missing, making it difficult to assess the statistical validity.

Round 2

Reviewer 2 Report

N/A

Reviewer 3 Report

Despite knowing that 1A8 treatment has limitations due to antigen masking, the authors did not validate the neutrophil depletion by flow cytometry (a standard method to assess depletion efficacy other than IHC) instead authors chose to provide additional information and clarifications by stating that they did not collect mice blood for further validation rather than address the issue experimentally (which they could have done by repeating the critical experiment with additional cell line as suggested in previous comments). Furthermore, I'm still perplexed as to why gMDSC or neutrophils were investigated in the first place. Although the author stated that gMDSC or neutrophils are frequently expanded in cancer patients and preclinical models, this does not explicitly justify their scientific premise to conduct this study without experimentally validating previously reported circulating and tissue-resident immune cells. Additionally, it is difficult to assess and validate the authors' claim of neutrophils in limb muscles of sham or isotype control KPC mice. As a result, as the author pointed out in the context of 4T1 cells, this concept and the results presented here are expected and do not represent novel research findings. Therefore, in my opinion, this study lacks the necessary novelty to be suitable in its present form for Cells.